# Cyclodextrin Complexes for the Treatment of Chagas Disease: A Literature Review

**DOI:** 10.3390/ijms25179511

**Published:** 2024-09-01

**Authors:** Fabrice Taio, Attilio Converti, Ádley Antonini Neves de Lima

**Affiliations:** 1Department of Pharmacy, Health Sciences Center, Federal University of Rio Grande do Norte, Natal 59012-570, Brazil; fabrice.taio.707@ufrn.edu.br; 2Department of Civil, Chemical and Environmental Engineering, Pole of Chemical Engineering, Genoa University, I-16145 Genoa, Italy; converti@unige.it

**Keywords:** cyclodextrins, Chagas disease, therapeutic activities, natural products

## Abstract

Cyclodextrins are ring-shaped sugars used as additives in medications to improve solubility, stability, and sensory characteristics. Despite being widespread, Chagas disease is neglected because of the limitations of available medications. This study aims to review the compounds used in the formation of inclusion complexes for the treatment of Chagas disease, analyzing the incorporated compounds and advancements in related studies. The databases consulted include Scielo, Scopus, ScienceDirect, PubMed, LILACS, and Embase. The keywords used were “cyclodextrin AND Chagas AND disease” and “cyclodextrin complex against *Trypanosoma cruzi*”. Additionally, a statistical analysis of studies on Chagas disease over the last five years was conducted, highlighting the importance of research in this area. This review focused on articles that emphasize how cyclodextrins can improve the bioavailability, therapeutic action, toxicity, and solubility of medications. Initially, 380 articles were identified with the keyword “cyclodextrin AND Chagas disease”; 356 were excluded for not being directly related to the topic, using the keyword “cyclodextrin complex against *Trypanosoma cruzi*”. Over the last five years, a total of 13,075 studies on Chagas disease treatment were found in our literature analysis. The studies also showed interest in molecules derived from natural products and vegetable oils. Research on cyclodextrins, particularly in the context of Chagas disease treatment, has advanced significantly, with studies highlighting the efficacy of molecules in cyclodextrin complexes and indicating promising advances in disease treatment.

## 1. Introduction

In 1891, Antoine Villiers made a pioneering discovery by identifying crystalline dextrins and isolating the oligosaccharides derived from starch using the enzyme cyclodextrin glycosyltransferase. This initial advance was crucial for understanding the complex processes of starch transformation. Thirty years later, Freudenberg and his team detailed the structure of cyclodextrins, providing a solid foundation for future research and development. The production of these molecules in their pure form was achieved in 1984, marking a significant milestone in the field. Since then, the study and application of cyclodextrins have expanded considerably, with numerous scientific publications and patents exploring their diverse applications. Cyclodextrins have demonstrated substantial potential in the pharmaceutical and food industries, being utilized in a wide range of products and technologies [1].

Chagas disease, caused by *Trypanosoma cruzi*, affects millions of people globally. The WHO aims to eradicate it as a public health concern by 2030, emphasizing the importance of early diagnosis. With 6 to 8 million infected and 75 million at risk, it is a major threat in Latin America. Migration has led to new cases in non-endemic countries. Without treatment, up to 30% of those infected may develop fatal chronic heart or digestive diseases [2]. The medications for Chagas disease, nifurtimox and benznidazole, were introduced over 50 years ago and are effective only in the early stages, with many side effects. New formulations and combinations have been tested, but none have surpassed benznidazole. Therefore, the discovery of new medications is a priority. The search for more effective and safer treatments is essential to improve disease management [3].

To address these challenges, cyclodextrins are used as encapsulants because of their ability to form reversible inclusion complexes with nonpolar molecules. The formation of these complexes is determined by the structural and physicochemical characteristics of both the compounds and the cyclodextrins. They form when the guest molecule fits into the cavity of the cyclodextrins [4,5].

Cyclodextrin-based treatment is currently limited to two nitro-heterocyclic drugs, namely, nifurtimox and benznidazole, which have been the main parasiticidal agents for almost five decades. However, it is important to highlight that the safety and efficacy profiles of these drugs are far from ideal [6,7]. Although the cure efficacy is relatively low, especially in the chronic phase of the disease, in the acute phase, the treatment achieves a cure rate of 76%, but in the chronic phase, this percentage drops to only 40%. Cyclodextrin chemotherapy faces challenges due to the specific toxicity of these compounds, the presence of sensitive or resistant strains of *Trypanosoma cruzi*, and the existence of cross-resistance among them [8]. Therefore, it is important to pave the way for research programs aimed at developing new drugs for Chagas disease [9].

The aim of this research is to conduct a literature review to identify the potential therapeutic activities of cyclodextrins that could be effectively used in combating Chagas disease. This review also aims to present alternative molecules reported in the literature that may contribute to and serve as guidance for the treatment of this disease, focusing on aspects such as therapeutic history, biological limitations, toxicity, and the type of cyclodextrin that may favor their encapsulation. These parameters, along with the information collected from clinical and experimental studies, are crucial for drawing conclusions about the effectiveness of incorporating these molecules into cyclodextrins. Thus, this review aims to provide a comprehensive overview of the therapeutic regimen and ongoing clinical trials, highlighting the potential use of cyclodextrins to overcome some therapeutic shortcomings.

## 2. Flowchart of This Study

For this systematic review, we used several databases, including Scopus, ScienceDirect, Scientific Electronic Library Online (Scielo), Latin American and Caribbean Literature in Health Sciences (LILACS), Embase, and the National Center for Biotechnology Information (PubMed). Following a comprehensive search, we proceeded to exclude articles that did not align with the central theme of this research, aiming to ensure consistency and relevance in our analysis. The results revealed that we found 23 articles in PubMed, 1 in Scielo, 1 in LILACS, 45 in Scopus, 45 in Embase, and 265 in Science Direct (Figure 1). After reviewing the abstracts of the initial 380 articles, we excluded 356 of them for not being directly related to the topic at hand. Our primary objective was to establish a solid foundation for the development of the topic, selecting articles that would allow us to understand the current state of research in this area and using similar articles as reference points for a more critical analysis. To organize the article selection process, we created a flowchart outlining the selection criteria based on relevant keywords, serving as a guide to achieving our research objectives.

The results of a rigorous filtering process, conducted on the topic of ‘cyclodextrin complexes against Trypanosoma cruzi’ and presented in Figure 1, led to the identification of 24 relevant publications. During the analysis, duplicates were eliminated, considering that some articles were published in different journals. After this step, 13 articles were deemed most pertinent, as detailed in Table 1. These articles provide the foundation for an in-depth discussion of the topic.

**Table 1 ijms-25-09511-t001:** Characterization of articles regarding specific studies on the inclusion of various compounds in cyclodextrins for the treatment of Chagas disease.

Year	Reference	Title	Goals	Conclusions
2023	[10]	New drug encapsulated incyclodextrin with promising anti-*Trypanosoma cruzi* activity.	Production and Characterization of a host–guest complex (Anti-Chagas Drug-Modified Chalcone (CHC) in 2-Hydroxypropyl-Beta-Cyclodextrin).	HPβCD/CHC showed promising activity against *Trypanosoma cruzi*. This complex offers improved water solubility and requires a lower amount of CHC to be effective.
2023	[11]	Elucidating the complexation of nifurtimox (NIF) with cyclodextrins.	Evaluate whether the formation of complexes with β-cyclodextrin and sulfobutyl ether-β-cyclodextrin would improve the solubility and dissolution rate of the drug.	β-CD/NIF and SBE-β-CD/NIF improved drug solubility and dissolution rate, showing significant stability in dissolution and crystallinity over 6 months at 25 °C and 40 °C.
2023	[12]	*O-allyl-lawsone* inclusion complex with 2-hydroxypropyl-β-cyclodextrin: Preparation, physical characterization, antiparasitic and antifunga activity.	Evaluate the antiparasitic and antifungal activity of *O-allyl-lawsone* (OAL) free and encapsulated in 2-hydroxypropyl-β-cyclodextrin (OAL MKN) against *Trypanosoma cruzi*.	HPβCD/OAL increased antiparasitic activity compared with the free form (OAL) while reducing cytotoxicity and enhancing selectivity for the trypomastigote form of *T. cruzi*.
2021	[13]	Characterization and trypanocidal activity of a drug carrier containing β-lapachone.	Investigate the in vitro action of anti-*T. cruzi*, effects of β-Lap encapsulated in 2-hydroxypropyl-β-cyclodextrin (2HP-β-CD), and its potential toxicity to mammalian cells.	The trypanocidal activity was increased by encapsulation of HP-β-CD/β-Lap compared with free naphthoquinone (β-Lap).
2020	[14]	Synthesis and biological evaluation of β-lapachone and nor-β-lapachone complexes with 2-hydroxypropyl-β-cyclodextrin as trypanocidal agents.	Study βLAP and its derivative complexes nor-β-Lapachone (NβL) with 2-hydroxypropyl-β-cyclodextrin to increase solubility and bioavailability.	HP-β-CD/βLAP and HP-β-CD/NβL increased the drug solubility and, additionally, vectorization was observed, resulting in higher biological activity against the epimastigote and trypomastigote forms of *T. cruzi*.
2022	[15]	Synthesis and study of the trypanocidal activity of catechol-containing 3-arylcoumarins, inclusion in β-cyclodextrin complexes and combination with benznidazole.	Evaluate trypanocidal activity and cytotoxicity of a series of catechol-containing 3-arylcoumarins, their combination with BZN, and inclusion in β-cyclodextrins (β-CDs).	Catechol-containing 3-arylcoumarins showed moderate trypanocidal activity against *Trypanosoma cruzi*, and their inclusion in β-cyclodextrins improved solubility. Combining these coumarins with benznidazole (BZN) further enhanced their effectiveness.
2018	[16]	Technological innovation strategies for the specific treatment of Chagas disease based on Benznidazole.	Conduct a literature review to identify current pharmaceutical technologies used in conjunction with BNZ to improve therapy for Chagas disease.	A lower concentration of BNZ was required to eliminate 50% of *T. cruzi* trypomastigote forms. This was achieved through the formation of BNZ/CD complexes and the modulation and targeting of anti-Chagas treatment using metal-organic frameworks.
2017	[17]	Benznidazole nanoformulates: A chance to improve therapeutics for Chagas disease.	Describe the characterization of several encapsulated formulations of benznidazole, currently a first-line medication for the treatment of Chagas disease.	The in vitro cytotoxicity of BZN/CDs was significantly lower than that of free benznidazole, while their trypanocidal activity was not impaired.
2011	[18]	Activity of a metronidazole analogue and its β-cyclodextrin complex against *Trypanosoma cruzi*.	Prepare an inclusion complex between a metronidazole iodide analog (MTZ-I) and cyclodextrin (CD) to develop a safer and more effective method of treating *Trypanosoma cruzi* infections.	MTZ-I and MTZ-I/β-CD were 10 times more active than MTZ, indicating that the presence of an iodine atom in the side chain increased trypanocidal activity while maintaining its cytotoxicity.
2011	[19]	Modulated dissolution rate of the antichagasic benznidazole inclusion complex and cyclodextrin using hydrophilic polymer.	Investigate the utility of hydroxypropylmethylcellulose (HPMC) polymer in controlling the release of BNZ from solid inclusion complexes with cyclodextrin to overcome the problem of its bioavailability.	The addition of HPMC to BZN/CD inclusion complexes significantly improved the dissolution rate and controlled drug release, showing promising potential for Chagas disease therapy.
2012	[20]	Benznidazole drug delivery by binary and multicomponent inclusion complexes using cyclodextrins and polymers.	Develop and characterize inclusion complexes in binary systems with BNZ and randomly methylated β-cyclodextrin (RMβCD), and in ternary systems with BNZ, RMβCD, and hydrophilic polymers.	Cyclodextrin-based inclusion complexes with benznidazole (BNZ) and hydrophilic polymers demonstrated effective, standardized, and safe drug delivery.
2008	[21]	Study of the interaction between hydroxymethyl nitrofurazone and 2-hydroxypropyl-β-cyclodextrin.	Characterize an NFOH inclusion complex in 2-hydroxypropyl-β-cyclodextrin (HP-β-CD).	HP-β-CD/NFOH significantly reduced the toxic effects of NFOH, according to preliminary toxicity studies and cell viability tests.
2007	[22]	Hydroxymethylnitrofurazone inclusion complex: dimethyl-β-cyclodextrin: a physicochemical characterization.	Characterize inclusion complexes formed between NFOH and dimethyl-β-cyclodextrin (DM-β-CD) through complexation/release kinetics and solubility isotherm experiments using ultraviolet (UV)–visible spectrophotometry and dynamics measurement.	NFOH/DM-β-CD showed improved solubility and favorable complexation, as demonstrated by solubility isotherm studies.

## 3. Brief Review

In this section, various elements related to cyclodextrin, the history of Chagas disease, and studies for its treatment are thoroughly explored.

### 3.1. Cyclodextrin

Natural cyclodextrins, specifically α-CDs, β-CDs, and γ-CDs, consist of sets of six, seven, and eight glucose units, respectively (Figure 2). These units are expressed differently, allowing for precise adjustment of both the potency and duration of medication action [18]. As for their synthetic counterparts, they can be categorized into three distinct classes as follows: hydrophilic, represented by 2-hydroxypropyl-β-CD (HP-β-CD); hydrophobic, exemplified by 2,6-di-O-ethyl-β-CD; and ionizable, with sulfobutyl ether β-CD (SBE-β-CD) serving as a prominent example. Cyclodextrin complexes are extensively utilized in pharmaceutical products, drug delivery systems, cosmetics, and various industries and are frequently incorporated into diverse pharmaceutical formulations [20]. Furthermore, they possess notable pharmacological properties, facilitating drug solubility, improving bioavailability, contributing to physicochemical stability, engaging in inclusion complex formation, and enhancing the sensory attributes of drugs, among other roles [23].

The pharmaceutical properties of cyclodextrins are attributed to their complex three-dimensional structure, which is a result of the abundant presence of hydroxyl groups, rendering them readily soluble in water. Cyclodextrins possess external cavities with a hydrophilic layer of water and internal cavities that repel water (hydrophobic) (Figure 3). These characteristics enable them to dissolve in aqueous solutions and encapsulate hydrophobic moieties within their cavities. The incorporation of “guest” molecules into cyclodextrin inclusion complexes in aqueous media serves as the basis for most applications in the pharmaceutical field. If the drug molecules have adequate size and properties suitable for the formation of inclusion complexes, a balanced equilibrium exists among free cyclodextrins, free drug molecules, and the inclusion complexes that form [24] (Figure 4).

Inside the cavity, there are two rings of C-H groups and a ring of glycosidic oxygen atoms, which impart hydrophobic characteristics to its internal structure. Table 2 provides a summary of the most significant physicochemical properties of natural cyclodextrins.

### 3.2. Industrial Applications

The cyclodextrin system is extensively explored and studied in various areas of the literature [26]. This includes fields such as food science, technology, and engineering [27,28,29], pharmaceutical industries [30,31,32,33], chemistry or chemical engineering [34,35], environmental sciences [36,37], and the cosmetics industry [38,39].

### 3.3. Complexation Mechanism

Cyclodextrins have the ability to form inclusion complexes in both liquid and solid media. After this complexation, they can enhance the physicochemical properties of the host substance. Cyclodextrins have the capacity to reduce the volatility of compounds, transform liquids into solids, mask the unpleasant odor and taste of certain compounds, prevent undesirable incompatibilities, improve bioavailability, and reinforce stability, even in the presence of light, heat, and oxidizing conditions. To achieve the formation of inclusion complexes with cyclodextrin, various methods are utilized, as each target substance to be encapsulated has its own particularities. The preparation of these complexes is a relatively simple process. The most common procedure involves agitating or mixing cyclodextrin in an aqueous solution that may vary in temperature (cold or hot) and pH (neutral, alkaline, or acidic), containing the molecules of the guest compound. Inclusion complexes can be acquired through a variety of techniques, including co-precipitation [40,41], co-evaporation [42,43,44,45], lyophilization and physical mixing [46], spray drying and kneading [47], and the use of supercritical fluid technology [48]. In the complexes (Figure 4), the host molecule, i.e., cyclodextrin, partially or entirely houses the guest molecule in its cavity. The formation of this complex is determined by the dimensional characteristics of both the cyclodextrin cavity and the guest molecule.

### 3.4. Chagas Disease

The complication of Chagas disease has been exacerbated because of recent population migrations in the Latin American continent [49]. These migrations involve the movement of people from rural to urban areas in Europe, North America [50], Japan, and Australia. Remarkably, the number of immigrants from North America to Europe doubled, from 910,402 in 2001 to over 2 million in 2004. Additionally, Japan, Australia, and Canada have experienced a significant influx of immigrants from Latin America in recent years [51]. Europe’s involvement is noteworthy, with the majority of Chagas disease cases being registered in countries such as Spain and Italy, followed by the United Kingdom, Portugal, Switzerland, France, and Sweden [52,53].

In 1909, the Brazilian doctor and scientist Carlos Ribeiro Justiniano das Chagas, as a researcher, managed to clarify and elucidate the contamination from the main vector to the host, the infecting forms, and the biological cycle. Chagas disease is caused by the flagellated protozoan *Trypanosoma cruzi*, which is an etiological agent. This parasite has an extensive biological cycle and goes through several evolutionary forms within the vertebrate host (man) and in insect vectors including triatomines, which are a family of insects in the order Hemiptera, suborder Heteroptera, family Reduviidae, popularly known as “kissing bugs” [54].

### 3.5. Biological Cycle

As shown in Figure 5, the biological cycle of *T. cruzi* begins with the penetration of the trypomastigote forms present in triatomine feces into the intact mucosa or damaged skin of the host after the insect has taken a blood meal. Then, the metacyclic trypomastigote forms penetrate the cells and differentiate into amastigotes. This form multiplies by binary division within the cell and differentiates again into trypomastigotes, which, through the rupture of the parasitized cell, are released into the circulating blood and can again infect another cell or be ingested by the insect. When the triatomine takes a blood meal in hosts infected by *T. cruzi*, it ingests the trypomastigote form, which in the insect’s midgut differentiates into the epimastigote form of the parasite and multiplies. Then, in the rectal ampulla of the vector, these forms differentiate again into trypomastigotes, which are again capable of infecting new hosts [55,56].

### 3.6. Nifurtimox and Benznidazole

In the 1970s, the following compounds were identified for the treatment of CD: nifurtimox and benznidazole [57], whose structures are depicted in Figure 6. These medications are used with the purpose of reducing both the duration and severity of the symptoms of the disease. The main benefits expected from treatment include reducing the presence of parasites in the body (parasitemia), preventing disease reactivation, increasing the patient’s life expectancy, and reducing clinical complications [9]. The first-line treatment is benznidazole, often chosen because of its safety and efficacy profile [58].

Nifurtimox works by generating oxygen, which leads to the intoxication of the parasite. Consequently, it can cause damage to the host’s tissues, thereby explaining the manifestation of several side effects. On the other hand, benznidazole does not cause oxidative damage in its antiparasitic action; instead, it blocks the production of proteins and RNA in both extracellular and intracellular forms found in the parasitized host [59]. Since the 1980s, the sale of nifurtimox has been suspended, initially in Brazil and later in other South American nations. The main drawback associated with both medications is their unwanted side effects (see Table 3) [59]. 

### 3.7. Studies Conducted on the Treatment of T. cruzi

After the acute phase, survivors enter the chronic phase of Chagas disease, often characterized by the absence of visible symptoms, which can persist indefinitely. Over time, patients in this phase may develop symptoms related to the cardiovascular, digestive, or both systems, resulting in cardiac and digestive forms of the disease [61]. Because of the disease’s prevalence in low-income populations, there has been little interest from the pharmaceutical industry in investing in treatments, as they do not represent significant financial returns. The complexity of the disease and limited resources allocated to research in this area has contributed to the scarcity of therapeutic advances over the decades. To date, no new drugs have been developed for the treatment of Chagas disease, as reported by the U.S. Food and Drug Administration. However, academic interest in the topic is significant, especially in public higher education institutions in Latin America [62]. To advance in this scenario, recent research has focused on the treatment of Chagas disease. Over the past five years, our analysis of the scientific literature revealed a total of 2655 studies in PubMed, 19 in Scielo, 144 in LILACS, 3106 in Embase, 5345 in ScienceDirect, and 1806 in Scopus (Figure 7).

## 4. Impact of Cyclodextrins on the Optimization of Drug Solubility and Efficacy

Despite the theoretical and practical importance of cyclodextrins and the extensive literature on them, their significant potential is still not well understood, requiring further research on their structure and properties. A promising area is the modification of standard cyclodextrins to enhance their complexation capacity with various substances, whether by inclusion in the cavity or binding to the external surface. Additionally, it is essential to investigate the physicochemical properties of the aqueous solutions of these systems to better understand the interactions between host and guest, as well as the inclusion complex with the aqueous solvent [63].

Cyclodextrins are chemically versatile and can be modified to produce mono- or polysubstituted derivatives, which can enhance their properties, such as solubility and stability, and adjust their complexation abilities. The process of complexation leads to significant changes in the spectral properties, reactivity, volatility, solubility, and stability of guest compounds. Thus, cyclodextrins hold great potential for application in various technological fields [64].

The structure of the inclusion complex that is formed depends on the drug and the cyclodextrins used, which can stabilize some drugs while destabilizing others. The effectiveness of stabilization is influenced by the types of cyclodextrins employed in the complexation. Although binary complexes of cyclodextrins and drugs can improve stability, cyclodextrins with low complexation efficiency may require higher concentrations to form these complexes. Adding a ternary agent to a binary complex can increase the stability constant and efficiency because of the synergistic interaction between the components. However, a third component that competes for inclusion in the cyclodextrin cavity can reduce overall efficiency, making the appropriate selection of ternary compounds crucial for the development of supramolecular systems. To achieve the desired drug stability, it is essential to identify which cyclodextrins provide the greatest stabilizing effect and determine the most suitable concentrations for each case [65].

## 5. Impact of Cyclodextrins with Benznidazole in the Treatment of Chagas Disease

Researchers are exploring new approaches for treating Chagas disease. Years of research have evolved with fresh perspectives on finding the best drug for *T. cruzi*. Thus, the research has focused on established drugs against Chagas disease, such as benznidazole. Soares-Sobrinho et al. [20] developed and characterized inclusion complexes with benznidazole and randomly methylated β-cyclodextrin (RMβCD), demonstrating that methylated cyclodextrins are a reliable and safe option for effective drug delivery. In the same year, Vinuesa et al. [17] developed nanoformulations of benznidazole in cyclodextrin, which reduced toxicity without compromising efficacy against *Trypanosoma cruzi*, indicating a potential improvement in Chagas disease treatment. Additionally, Ferraz et al. [16] highlighted the efficacy of benznidazole at low concentrations, while Sá-Barreto et al. [19] proposed BNZ-HPβCD to enhance drug bioavailability. Finally, Pozo-Martínez et al. [15] observed that flavonoids combined with benznidazole in complexes with β-cyclodextrins increased drug solubility. These findings represent promising advances in the search for more effective and safer treatments for Chagas disease.

These discoveries signify promising strides towards more effective and safer treatments for Chagas disease. The evolution of studies with benznidazole has demonstrated significant progress over the years, with the development of inclusion complexes and nanoformulations that enhance efficacy and reduce toxicity. Cyclodextrins, especially β-cyclodextrins and their derivatives, have proven to be valuable tools in improving the solubility and bioavailability of benznidazole, offering new opportunities for more effective and safer treatments against *Trypanosoma cruzi*.

## 6. Impact of Cyclodextrins with Natural Products in the Treatment of Chagas Disease

Research in the field of Chagas disease treatment has advanced significantly, overcoming the limitations of existing medications and exploring new approaches. There is a growing emphasis on the use of β-cyclodextrins and their derivatives (Table 4), as well as natural molecules (Table 5), to develop more effective formulations. Comparative studies have shown that metronidazole iodide is 10 times more potent against *Trypanosoma cruzi*, maintaining cytotoxicity when complexed with β-cyclodextrin [19]. In 2023, Bedogni et al. [11] investigated the complexation of nifurtimox with cyclodextrins to enhance the drug’s solubility and dissolution, achieving a stable complex. Nicoletti et al. [12] found increased antiparasitic and antifungal activity of O-allyl-lawsone (OAL) when complexed with β-cyclodextrin. Zanetti et al. [10] highlighted the antiparasitic activity of chalcone complexes with hydroxypropyl β-cyclodextrin, along with improved water solubility. Over the years, cyclodextrins have emerged as crucial tools for overcoming the limitations faced by various compounds [66]. Nitrofurazone, initially active against Gram-positive microorganisms, later demonstrated efficacy against *Trypanosoma cruzi* [67]. Studies involving cyclodextrin inclusion complexes, such as DM-β-CD and HP-β-CD, have shown increased solubility and reduced toxicity [21,22]. β-lapachone, derived from natural products and chemically known as 3,4-dihydro-2,2-dimethyl-2H-naphtho[1,2-b]pyran-5,6-dione, belonging to the Bignoniaceae family and commonly found in Brazil [68], has also been studied in relation to *Trypanosoma cruzi* [69]. Complexes formed with 2-hydroxypropyl-β-cyclodextrin increased the drug’s solubility, resulting in enhanced trypanocidal activity [13,14]. Cyclodextrin complexes with various drugs have demonstrated significant improvements in solubility, stability, and antiparasitic activity against *Trypanosoma cruzi* (*T. cruzi*), often resulting in reduced cytotoxicity and safer treatments. The combination of benznidazole (BZN) with cyclodextrins and hydrophilic polymers has enhanced both the efficacy and safety of treatments, presenting a promising approach for Chagas disease and other parasitic infections. Cyclodextrins such as HP-β-CD and SBE-β-CD have proven particularly effective in improving drug bioavailability and enhancing antiparasitic activities by forming stable complexes and reducing toxicity. The data in Table 1, Table 4 and Table 5 indicate that β-cyclodextrin derivatives are crucial in this context as they increase drug solubility and enhance therapeutic efficacy. Additionally, the incorporation of polymers and other agents can further amplify therapeutic effects and minimize side effects. The significance of cyclodextrins in developing new treatments for Chagas disease is evident, as they optimize pharmacological properties and reduce drug toxicity. These advances highlight the ongoing relevance of research into cyclodextrins and natural products, promoting innovations that have the potential to transform therapeutic strategies for Chagas disease.

## 7. Discussion

Nowadays, the use of natural products in complexation with cyclodextrins represents a promising and relevant strategy in pharmaceutical research. Research involving cyclodextrins with natural products aims not only to enhance the efficacy of bioactive molecules but also to explore their therapeutic properties in a safer and more effective manner [80]. The example of thymol, a natural compound with various beneficial properties such as antioxidant, antiseptic, and antifungal [81], illustrates how complexation with cyclodextrins like γ-cyclodextrin can enhance its applications, as demonstrated in studies by Zhang Y. and colleagues [82]. Additionally, the application of HP-β-cyclodextrin by Serna-Escolano et al. [83] as an alternative to synthetic fungicides highlights the potential of cyclodextrins in food preservation and protection against diseases.

Naphthoquinones are abundant natural products derived from naphthalene that constitute the quinone group fused to a benzene ring [84] and exhibit activity against fungi [68] and Chagas disease [85]. Studies on naphthoquinones, such as the one conducted by Oliveira da Silva and colleagues [86], underscore the growing importance of exploring natural compounds for treatments against diseases such as Chagas disease and viral infections like SARS-CoV-2. The complexation ability of β-cyclodextrin and its methylated derivatives proves crucial not only for improving the solubility and bioavailability of these compounds but also for enhancing their pharmacological properties.

Thus, the use of natural products in combination with cyclodextrins not only expands therapeutic possibilities but also represents a significant step towards more effective and safer treatments, contributing to ongoing advancements in research and therapeutic innovation.

Many studies indicate that the inclusion of cyclodextrins is widely used in the complexation of plant-derived molecules, especially terpenes, resinous oils, and essential oils, aiming to enhance their chemical and pharmacological properties [10,87]. Encapsulated vegetable oils in carriers have shown effectiveness in improving physical stability under light and heat conditions, increasing bioavailability, and reducing unpleasant aromas and flavors, as well as decreasing volatility and toxicity. Thus, molecular complexation with cyclodextrin emerges as a promising technique for encapsulating these oils [88,89].

Copaiba oil, extracted from the tree of the same name common in the Amazon region of Brazil [90], is recognized for its antimicrobial, healing, analgesic, antitumor, anti-inflammatory, and *Trypanosoma cruzi* properties [91,92,93,94,95,96]. Biologically, caryophyllenes are responsible for the recognized therapeutic properties of copaiba oil [97]. Components present in copaiba oils face challenges such as degradation, oxidation, storage, unpleasant taste and odor, and low water miscibility. To address these challenges effectively, oil complexation with cyclodextrins emerges as a practical solution. In addition to overcoming these limitations, complexation can offer a way to produce solid materials from the complexation, potentially yielding derivatives with improved solubility, stability, and better taste [95]. However, toxicity varies according to the route of administration, with the use of parenteral and intravenous routes discouraged because of the risk of nephrotoxicity [98,99].

This focus on the use of vegetable oils complexed with cyclodextrins not only demonstrates promising advances in research but also highlights the potential of these formulations as innovative therapeutic alternatives for treating various diseases, such as Chagas disease.

## 8. Final Considerations

It is worth noting that a large number of in-depth studies have been published on complexation with cyclodextrins since the beginning, mainly focused on the two drug alternatives that existed at the time, nifurtimox and benznidazole, which revealed they were more active in the acute phase of Chagas disease [100]. Benznidazole has limited solubility and low permeability [101]. Thus, cyclodextrin was mixed with benznidazole to achieve the best results in terms of efficacy and tolerance [102]. All articles involving studies with modified, combined, or synthesized benznidazole along with cyclodextrins showed positive results in terms of bioavailability, solubility, and low toxicity [17,18,19,20]. Nevertheless, benznidazole is not recommended, nor is it the choice in the chronic phase, with practically no effect [103]. Although various efforts have been made to seek new therapeutic targets, the treatment of Chagas disease continues without an effective solution. Therefore, derivatives or drug base families (nifurtimox and benznidazole) and natural products from plants with antiprotozoal [104] and antibacterial properties are being explored along with the technique of inclusion complexation. These are potent, thanks to complexation, and have the potential for therapeutic effect in any stage of the disease [105]. Unfortunately, many studies have stopped at in vitro tests, and few have progressed to in vivo tests [104].

## 9. Conclusions

Research on cyclodextrins has witnessed remarkable growth in recent years, expanding into a wide range of disciplines and finding applications in various innovative technologies. This progress highlights the potential of these molecules to address complex challenges across multiple fields, including medicine. Specifically, in the context of Chagas disease treatment, cyclodextrins have emerged as a promising solution for enhancing drug bioavailability, a critical factor in increasing therapeutic efficacy. This is attributed to their favorable mechanism of action against the *Trypanosoma cruzi* parasite, the causative agent of the disease.

Between 2007 and 2023, a series of in-depth studies have underscored the importance of incorporating both natural and synthetic drugs into cyclodextrin complexes. These studies have shown promising results, providing evidence that complexation with cyclodextrins can significantly improve the physicochemical properties of drugs, such as solubility and stability, while also enhancing their biological activities against *T. cruzi*. Such findings suggest that these complexes could represent a substantial advancement in the fight against Chagas disease, particularly in critical phases of treatment where drug efficacy is limited.

Given the severity of Chagas disease and the urgent need for more effective therapies, it is crucial that research continues, with a special focus on preclinical phases. These additional studies are essential for refining our understanding of the interactions between cyclodextrins and drugs and for developing therapeutic approaches that could eventually be scaled up to combat this devastating disease. The continuation and deepening of this research could pave the way for more effective, safe, and accessible treatments, offering hope to millions of people affected by Chagas disease.

## Figures and Tables

**Figure 1 ijms-25-09511-f001:**
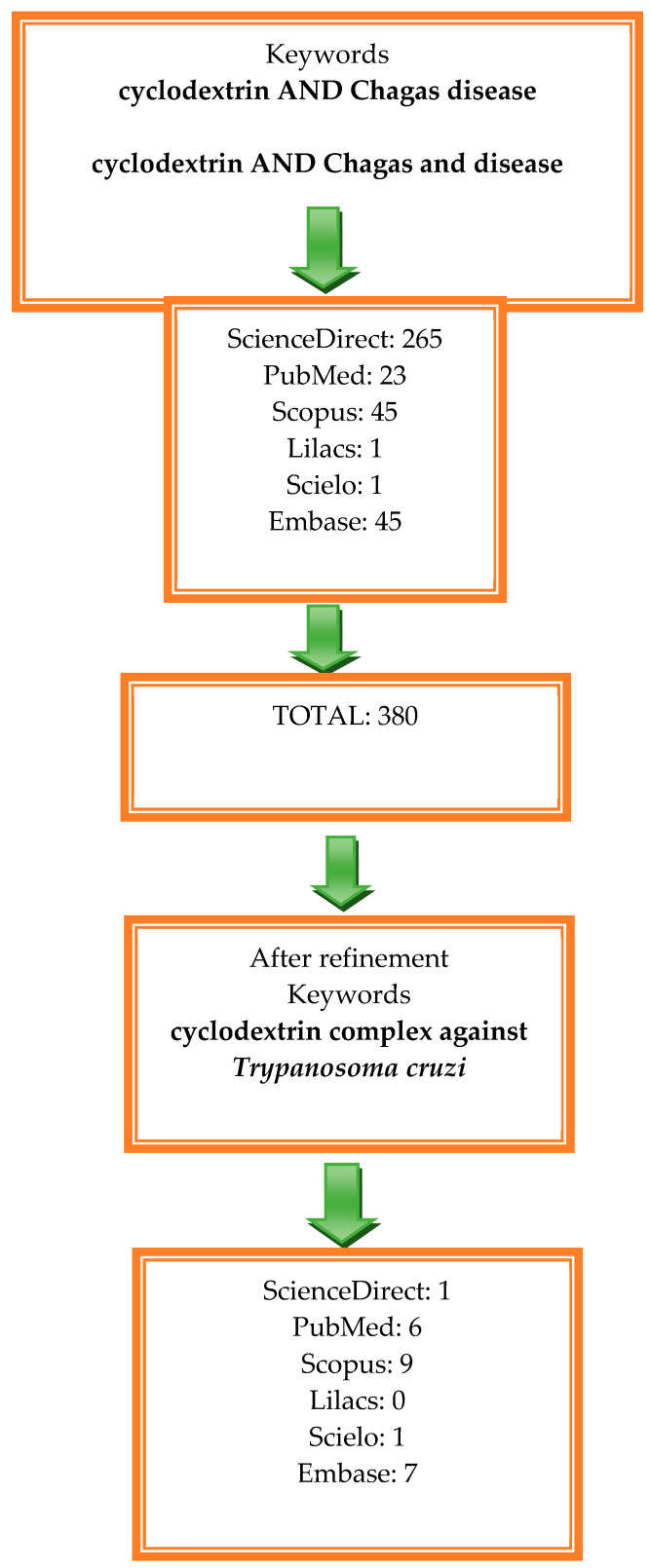
Flowchart illustrating the methodology employed for article selection.

**Figure 2 ijms-25-09511-f002:**
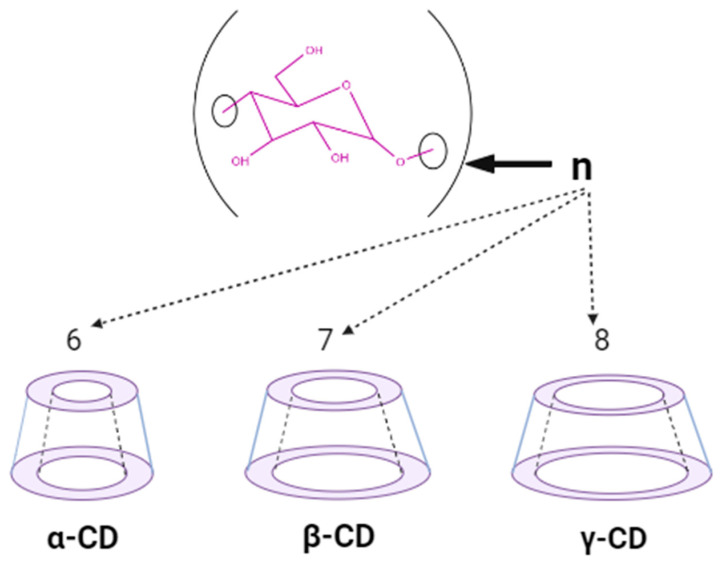
The format of natural cyclodextrins according to the amount of the glycosidic group.

**Figure 3 ijms-25-09511-f003:**
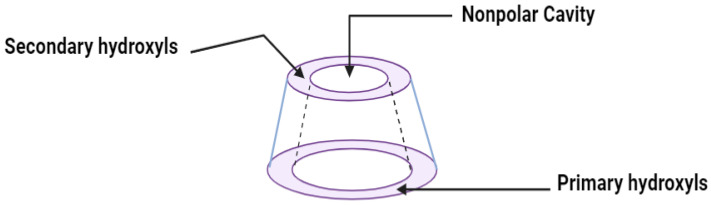
The external and internal cavities of cyclodextrins.

**Figure 4 ijms-25-09511-f004:**
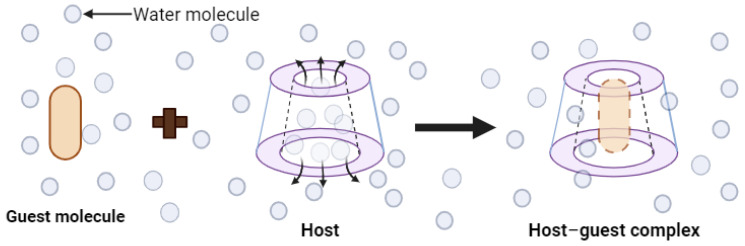
Schematic representation of the formation of an inclusion complex with cyclodextrins.

**Figure 5 ijms-25-09511-f005:**
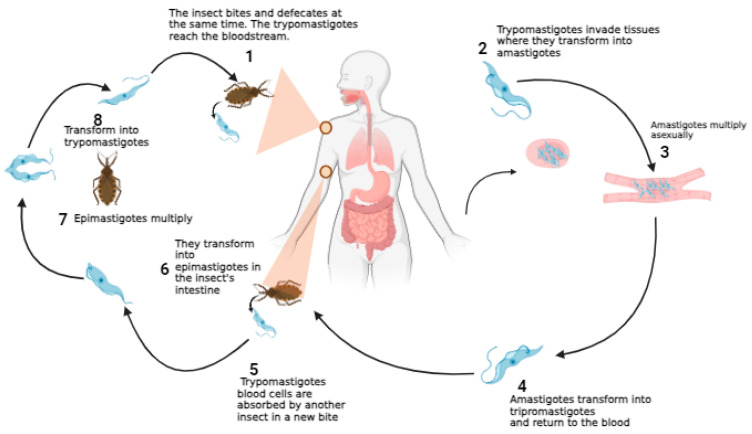
The biological cycle of Chagas disease in the human host.

**Figure 6 ijms-25-09511-f006:**
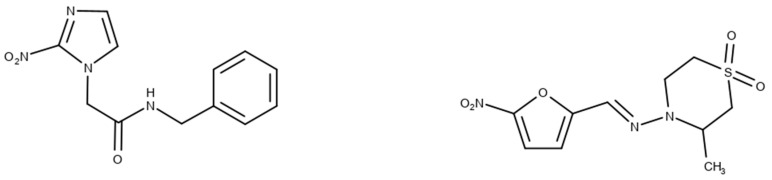
Benznidazole (**on the left**) and Nifurtimox (**on the right**) drugs for the treatment of Chagas disease.

**Figure 7 ijms-25-09511-f007:**
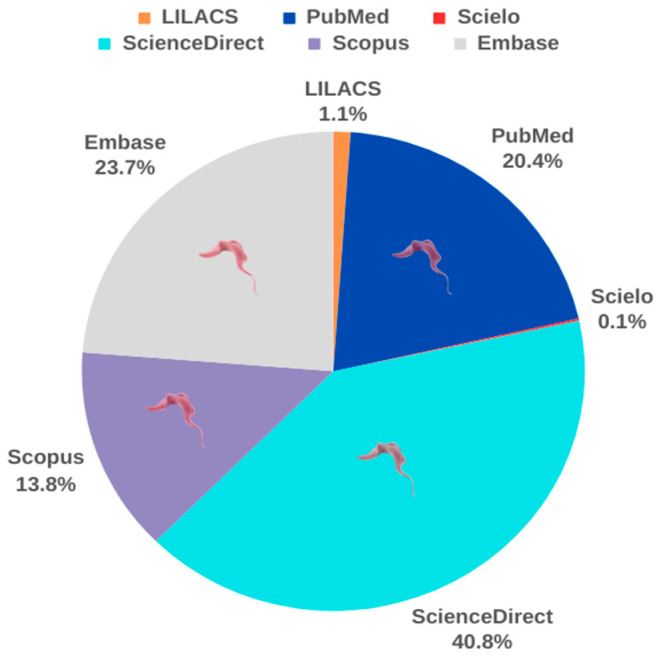
Evolution of the number of scientific publications on Chagas disease in the last 5 years.

**Table 2 ijms-25-09511-t002:** Physicochemical properties of natural cyclodextrins [25].

Cyclodextrin	Glucose Unit Number	Molecular Weight	Cavity Diameter (Å)	Cavity Volume (Å^3^)	Aqueous Solubility at 25° C (% *m*/*v*)
α-CD	6	972	4.5–5.3	174	14.5
β-CD	7	1135	6.0–6.5	262	1.85
γ-CD	8	1297	7.5–8.3	427	23.2

**Table 3 ijms-25-09511-t003:** Side effects observed in the specific treatment of Chagas disease [60].

Symptom/Sign	Benznidazol	Nifurtimox
Anorexia	++	+++
Headache	+	++
Dermatopathy	+++	+
Psychic excitement	-	+++
Gastralgia	+	+++
Insomnia	+	++
Nausea	++	+++
Weight loss	+	+++
Polyneuropathy	+	+++
Vomiting	++	+++

Legend: - (no effect), + (almost no effect), ++ (moderate effect), +++ (pronounced effect).

**Table 4 ijms-25-09511-t004:** Structures of the cyclodextrins most used to treat Chagas disease.

Type of Cyclodextrin	Reference	Cyclodextrin Structure
β-cyclodextrin (β-CD)	[70]	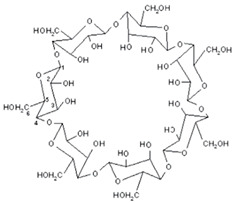
2-Hydroxypropyl-β-cyclodextrin (HP-β-CD)	[71]	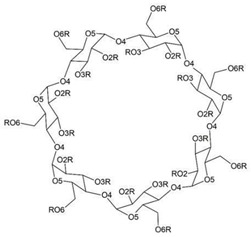
Dimethyl-β-cyclodextrin (DM-β-CD)	[72]	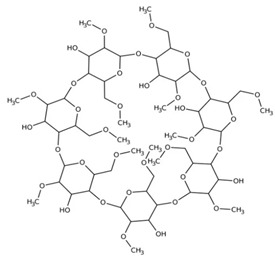
Ether sulfobutílico-β-cyclodextrin (SBE-β-CD)	[73]	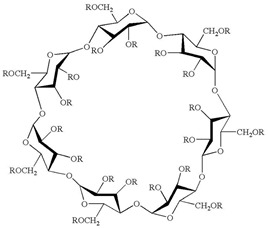

**Table 5 ijms-25-09511-t005:** Structures of the drugs tested to carry out the inclusion complex against *T. cruzi*.

Molecule Name	Reference	Molecular Structure
HidroximetilNitrofurazone (NFOH)	[74]	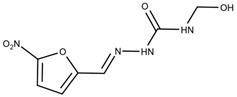
Chalcones (CHC)	[75]	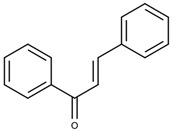
*O*-allyl-lawsone (OAL)	[12]	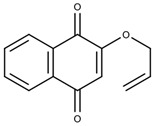
Nifurtimox (NF)	[76]	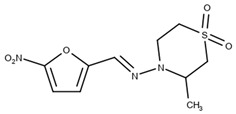
Benznidazole (BNZ)	[76]	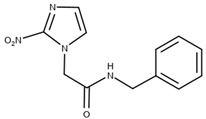
Metronidazole (MTZ)	[77]	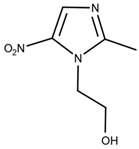
β-lapachone (β-Lap)	[78]	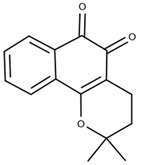
Nor-β-lapachone (NβL)	[79]	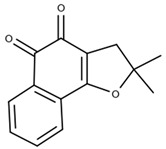

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
