# Peer review of "Cyclodextrin Complexes for the Treatment of Chagas Disease: A Literature Review"

_ijms, 2024, doi:10.3390/ijms25179511_

Round 1

Reviewer 1 Report

Comments and Suggestions for Authors

The review focuses on 18 articles highlighting how cyclodextrins can improve drug bioavailability, therapeutic action, toxicity, and solubility for active principles used against Chagas diseases. The topic may be interesting for the readers of IJMS but should be improved before publication.

Add a paragraph to Introduction on the nature and symptoms of Chagas disease.

What does it mean exactly „composition capability”?

Try to use the same abbreviation for the same things (b-Lap and bLAP, 2-HP-b-CD and HP-b-CD and HPbCD) and different ones to different things (CD for Chagas disease and for cyclodextrin)

Fig. 3 shows external (?) and internal cavities. External cavities are not on the drawing. What do authors mean on external cavities?

All the complex forming technologies start with mixing the host and guest either dissolved or in solid state in the presence of different amount of solvent, usually water and then the solvents are removed. Delete physical mixing as it is part of each technology mentioned here (L. 194).

Are metronidazole iodide, nifurtimox and the other compounds discussed in part 5 really natural products? Give their source in the nature.

Table 4 and 5 are not cited in the text.

Table 4 contains some text not written in English. Use chemical formulas in the same style.

The sentence in L.381–385 is not clear. Try to rephrase it.

The conclusion says: “favorable mechanism of action (of CDs) against T. cruzi”. But it was not discussed in the manuscript. It is well known that CDs themselves have antibacterial effect (e.g. against influenza and HIV), but in the manuscript only the complexes were discussed. The antibacterial effect against T. cruzi might not have been studied so far.

The reference list needs some editing.

Some typos:

Abstract: chags (?)

L. 137 disease

Table 1 hydroxymethyl (space) nitrofurazone

Table 5 hydroxymethyl (spelling)

L. 355 86–91 (not 86,91)

Author Response

Answers to the suggestions of reviewer 1

  1. Add a paragraph to the introduction about the nature and symptoms of Chagas disease.

Response to suggestion 1: The nature and symptoms of Chagas disease are already covered in Section 3.4, Chagas Disease.

  1. Figure 3 shows external (?) and internal cavities. External cavities are not depicted in the drawing. What do the authors mean by external cavities?

Response to suggestion 2: When we refer to the external cavity, we are talking about the panoramic view of the cyclodextrin structure, which allows for observation of both the interior and exterior of the molecule. This enables us to see the shape of the cyclodextrin inside and outside.

  1. Are metronidazole iodide, nifurtimox, and the other compounds discussed in Section 5 truly natural products? Please provide their sources in nature.

Response to suggestion 3: Metronidazole iodide and nifurtimox are compounds that have been associated with substances of natural origin (derived or not), such as O-allyl-lawson (OAL), nitrofurazone, and β-lapachone. The sources of this information are presented in the manuscript.

The other considerations will be corrected.

Reviewer 2 Report

Comments and Suggestions for Authors

1.                      The introduction could be improved by integrating a more detailed discussion of the current treatment challenges for Chagas disease and the potential role of cyclodextrins in overcoming these challenges.

2.                      The discussion on the impact of cyclodextrins in improving drug solubility, bioavailability, and therapeutic efficacy is thorough and well-supported by the literature. However, the article could benefit from a more critical analysis of the limitations and challenges associated with cyclodextrin-based therapies. For example, potential issues such as the stability of cyclodextrin complexes, the cost of production, and regulatory hurdles are not sufficiently addressed.

3.                      The formatting of the references is inconsistent in some places. Ensuring uniform citation style throughout would enhance the professional appearance of the article.

4.                      Discuss the practical implications of the findings for clinical practice, including how cyclodextrin complexes could be integrated into existing treatment protocols.

5.                      Address the economic aspects of cyclodextrin use, such as cost-effectiveness and potential financial barriers to implementation.

6.                      Offer a more detailed perspective on future research directions, including specific areas where further investigation is needed, such as long-term safety studies and clinical trials.

Comments on the Quality of English Language

Author Response

Answers to the suggestions of reviewer 2

  1. Address the economic aspects of cyclodextrin use, such as cost-effectiveness and potential financial barriers to implementation.

Response to suggestion 1: The effectiveness and cost of cyclodextrins are not fully addressed in this study because many articles already utilized beta-cyclodextrins due to their low cost and accessibility. The cyclodextrin range offers significant advantages due to its larger ring size, but it is more challenging to obtain due to its high cost, as reported in several articles. The focus of this study is indeed the application of cyclodextrins in the treatment of Chagas Disease, using molecules of therapeutic interest.

  1. Provide a more detailed perspective on future research directions, including specific areas where further investigation is needed, such as long-term safety studies and clinical trials.

Response to suggestion 2: As mentioned, a more detailed analysis of future research directions should focus exclusively on Chagas Disease, with an emphasis on improving and continuing studies up to the in vitro phase, which is currently often overlooked.

The other considerations will be corrected.

Reviewer 3 Report

Comments and Suggestions for Authors

Comments are in the attached file.

Comments on the Quality of English Language

Minor editing of english language is required.

Author Response

Answers to the suggestions of reviewer 3

  1. The presented abstract is poorly arranged, it should be revised, namely its background, relevance and/or originality of the review, and further investigations. It is very unclear and vague.

Response to suggestion 1:  The summary specifically targets the content that was detailed

  1. Lines 51-53: Please explain and justify this statement in the text.

Response to suggestion 2: The justification of this text is in [3,4,5].

  1. Table 1: The data included is unclear and vague. What results are the conclusions presented based on? What was tested and against what? Please rearrange and complete with suitable data accordingly to sustain the conclusions shown in Table 1.

Response to suggestion 3: The table is organized in such a way that each title leads to a goal, which in turn shows the conclusions. Each conclusion reveals the results of the anti-T. cruzi activities observed according to the type of inclusion complex formed.

  1. Table 3: Please explain in legend the significance of the exhibited signs: ‘+’. ‘++’ and ‘+++’.

Response to suggestion 4: Legend: - (no effect), + (almost no effect), ++ (moderate effect), +++ (pronounced effect). This legend has been added at the bottom of the table.

  1. Figures 2, 3 and 4 could be combined, there is no need to be separated.

Response to suggestion 5: We cannot combine everything into a single topic. The idea of the brief summary is to describe what cyclodextrins are, explain how they are internally structured by illustrating a molecule with its parts, and finally, show the complexation mechanism by illustrating a molecule being incorporated into the polymer. Following that, a literature update.

  1. Table 4 is irrelevant, please remove it.

Response to suggestion 6: It is important to keep this table, as recently the most commonly used cyclodextrins for forming complexes have been of the beta type. This may prompt interest in exploring other types of cyclodextrins that could potentially offer better therapeutic outcomes in the future.

  1. Sections 3.6, 3.7, 4, and 5 are very poorly developed and request a deep revision. It will be needed more consolidated information according to the data presented in Table 1.

Response to suggestion 7: Sections 3.6 and 3.7 are not related to Table 1. They are brief reviews on Chagas disease (3.6) and statistical studies on research conducted in the last 5 years on Chagas disease, regardless of type and not necessarily involving cyclodextrins. This serves to show that the disease remains of interest up to the present day. Sections 4 and 5 have undergone improvements to satisfy this suggestion.

  1. Please update and revise older references: 1, 2, 6, 18, 22, 23, 35, 36, 39, 45, 52, 56, 57, 61, 63, 65, 88, and 89.

Response to suggestion 8: The issue of reference updates was addressed in some cases, but in others, it was not possible, as we need to refer to the main author to maintain consistency with what we intend to address in the article. Nonetheless, recent articles from 2023-2024 have been added in some sections.

The other considerations will be corrected.

Round 2

Reviewer 2 Report

Comments and Suggestions for Authors

The manuscript has been improved, enough for being accepted.

Author Response

Reviewr 2 in Round 1:   Comments and Suggestions for Authors

The manuscript has been improved, enough for being accepted.

Reviewer 3 Report

Comments and Suggestions for Authors

Manuscript

Cyclodextrin Complexes for the Treatment of Chagas Disease: A Literature Review

Reviewer: Comments and Suggestions for Authors

2nd revision

Thank you very much for the corrections and answers to my comments in the first review. Overall they meet my expectations. However, I have some elements that have to be checked:

-        Please uniform genus and species names in italics.

-        Language should be revised. There are some typos errors.

-        The manuscript still has in the section of references papers that could be updated, as indicated in the last revision (namely [1] and [2]). Please revise and corroborate the list of references accordingly.

-        Please, as previously mentioned in the first revision, rearrange and complete with suitable data the conclusions shown in Table 1, it is very inadequate and not provide sufficient data to readers.

-        Please integrate the information in tables 1, 4 and 5 in the text. Tables are summaries, or information that corroborates the data presented in the text, and in this context, they are dissociated and/or not properly addressed and justified in the manuscript.

Comments on the Quality of English Language

Minor editing of English and language is required.

Author Response

Reviewer 3:

-         Please, as previously mentioned in the first revision, rearrange and complete with suitable data the conclusions shown in Table 1, it is very inadequate and not provide sufficient data to readers.

Reponse : The table has been corrected once again to provide readers with a clearer understanding, considering that (Drug/CD) indicates the specific complex that was studied.

-         Please integrate the information in tables 1, 4 and 5 in the text. Tables are summaries, or information that corroborates the data presented in the text, and in this context, they are dissociated and/or not properly addressed and justified in the manuscript.

Reponse : This section has already been covered in "6. Impact of Cyclodextrins with Natural Products in the Treatment of Chagas Disease." Nevertheless, we have added something at the end to address the complaint.

The other questions have been addressed.